# Multi-Objective Bayesian Optimization over High-Dimensional Search Spaces

**Samuel Daulton**[*,1,2]        **David Eriksson**[*,2]        **Maximillian Balandat**[2]        **Eytan Bakshy**[2]

[*]Equal contribution
[1]University of Oxford, Oxford, UK
[2]Meta, Menlo Park, USA

## Abstract

Many real world scientific and industrial applications require optimizing multiple competing black-box objectives. When the objectives are expensive-to-evaluate, multi-objective Bayesian optimization (BO) is a popular approach because of its high sample efficiency. However, even with recent methodological advances, most existing multi-objective BO methods perform poorly on search spaces with more than a few dozen parameters and rely on global surrogate models that scale cubically with the number of observations. In this work we propose MORBO, a scalable method for multi-objective BO over high-dimensional search spaces. MORBO identifies diverse globally optimal solutions by performing BO in multiple local regions of the design space in parallel using a coordinated strategy. We show that MORBO significantly advances the state-of-the-art in sample efficiency for several high-dimensional synthetic problems and real world applications, including an optical display design problem and a vehicle design problem with 146 and 222 parameters, respectively. On these problems, where existing BO algorithms fail to scale and perform well, MORBO provides practitioners with order-of-magnitude improvements in sample efficiency over the current approach.

## 1 INTRODUCTION

The challenge of identifying optimal trade-offs between multiple complex objective functions is pervasive in many fields, including machine learning [Sener and Koltun, 2018], science [Gopakumar et al., 2018], and engineering [Marler and Arora, 2004, Mathern et al., 2021]. For instance, Mazda recently proposed a vehicle design problem in which the goal is to optimize the widths of 222 structural parts in order to minimize the total weight of three different vehicles while simultaneously maximizing the number of common gauge parts [Kohira et al., 2018]. Additionally, this problem has 54 black-box constraints that enforce important performance requirements such as collision safety. Evaluating a design requires either crash-testing a physical prototype or running computationally demanding simulations. In fact, the original problem was solved on what at the time was the world's fastest supercomputer and took around 3,000 CPU years to compute [Oyama et al., 2017]. Another example is designing optical components for AR/VR applications, which requires optimizing complex geometries described by hundreds of parameters in order to identify designs that yield optimal trade-offs between image quality and efficiency of the optical device. Evaluating a design involves either fabricating and measuring prototypes or running computationally intensive simulations. For such problems, sample-efficient optimization is paramount.

Bayesian optimization (BO) has emerged as an effective, general, and sample-efficient approach for "black-box" optimization [Jones et al., 1998] and is highly effective for machine learning hyperparameter tuning [Turner et al., 2021]. However, in its basic form, BO is subject to important limitations. In particular, (i) successful applications typically consider low-dimensional search spaces, usually with less than 20 tunable parameters [Frazier, 2018], (ii) inference with the typical Gaussian Process (GP) surrogate models incurs cubic time complexity with respect to the number of data points, which prevents usage in the large-sample regime that is often necessary for high-dimensional problems, and (iii) most methods focus on single objective unconstrained problems. As a result, BO cannot easily be applied to either of the aforementioned Mazda vehicle design or the AR/VR optical design problems. Moreover, high dimensional multi-objective problems requiring sample-efficient optimization are prevalent in many real-world settings such as groundwater remediation [Akhtar and Shoemaker, 2015], cell network configuration [Dreifuerst et al., 2021], and water resource management [Bai et al., 2017]. The state-of-

*Accepted for the 38*[th] *Conference on Uncertainty in Artificial Intelligence* (UAI 2022).

the-art approach for this class of problems is NSGA-II [Deb et al., 2002], a popular evolutionary strategy, but with poor sample-efficiency, which hinders the progress of the scientists running these experiments.

In this paper, we close this gap by making BO applicable to challenging high-dimensional multi-objective problems. To do so, we propose an algorithm called MORBO ("Multi-Objective Regionalized Bayesian Optimization") that optimizes diverse parts of the global Pareto frontier in parallel using a coordinated set of local trust regions (TRs). As shown in Figure 1 (left), TRs are located at different solutions with diverse trade-offs between objectives. MORBO performs local BO in each TR to mitigate over-exploration, a phenomenon that plagues many algorithms in high-dimensional settings [Eriksson and Poloczek, 2021]. To enable scaling to large evaluation budgets, MORBO leverages *local* GP surrogate models of the objective function, which reduces the time complexity for GP inference from $O(n^3)$, where $n$ is the number of data points, to $O(n_{\mathcal{T}}^3)$, where $n_{\mathcal{T}} \ll n$ is the number of local data points for a TR $\mathcal{T}$. To facilitate efficient and collaborative global optimization, MORBO *passes information* between TRs in the following two ways: (1) Observations collected by one TR are shared with the others—which is particularly important when the TRs overlap as shown in Figure 1, (2) MORBO selects a batch of candidates by leveraging the TRs to collaboratively maximize a global utility. To ensure efficient global optimization, MORBO terminates under-performing TRs and allocates new TRs according to a global policy with a theoretical performance guarantee—a property that sets MORBO apart from most existing methods.

The significance of MORBO is that it is the *first* multi-objective BO method that scales to hundreds of tunable parameters and thousands of evaluations, a setting where practitioners have previously had to fall back on alternative methods with much lower sample-efficiency, such as NSGA-II. Our comprehensive evaluation demonstrates that MORBO yields *order-of-magnitude* savings in terms of time and resources compared to state-of-the-art methods on challenging high-dimensional multi-objective problems.

## 2 BACKGROUND

### 2.1 PRELIMINARIES

#### 2.1.1 Multi-Objective Optimization

In multi-objective optimization (MOO), the goal is to maximize (without loss of generality) a vector-valued objective function $\boldsymbol{f}(\boldsymbol{x}) = [f^{(1)}(\boldsymbol{x}), ..., f^{(M)}(\boldsymbol{x})] \in \mathbb{R}^M$, where $M \geq 2$ while satisfying black-box constraints $\boldsymbol{g}(\boldsymbol{x}) \geq \boldsymbol{0} \in \mathbb{R}^V$ where $V \geq 0$, $\boldsymbol{x} \in \mathcal{X} \subset \mathbb{R}^d$, and $\mathcal{X}$ is a compact set. Usually, there is no single solution $\boldsymbol{x}^*$ that simultaneously maximizes all $M$ objectives and satisfies all $V$ constraints.

Hence, objective vectors are compared using Pareto domination.

**Definition 2.1.** An objective vector $\boldsymbol{f}(\boldsymbol{x})$ *Pareto-dominates* $\boldsymbol{f}(\boldsymbol{x}')$, denoted as $\boldsymbol{f}(\boldsymbol{x}) \succ \boldsymbol{f}(\boldsymbol{x}')$, if $f^{(m)}(\boldsymbol{x}) \geq f^{(m)}(\boldsymbol{x}')$ for all $m = 1, ..., M$ and there exists at least one $m \in \{1, ..., M\}$ such that $f^{(m)}(\boldsymbol{x}) > f^{(m)}(\boldsymbol{x}')$.

**Definition 2.2.** The *Pareto frontier* (PF) is the set of optimal trade-offs $\mathcal{P}(X)$ over a set of designs $X \subseteq \mathcal{X}$:

$$\mathcal{P}(X) = \{\boldsymbol{f}(\boldsymbol{x}) : \boldsymbol{x} \in X, \nexists\, \boldsymbol{x}' \in X \; s.t. \; \boldsymbol{f}(\boldsymbol{x}') \succ \boldsymbol{f}(\boldsymbol{x})\}$$

Under black-box constraints, the *feasible Pareto frontier* is defined as $\mathcal{P}_{\text{feas}}(X) = \mathcal{P}(\{\boldsymbol{x} \in X : \boldsymbol{g}(\boldsymbol{x}) \geq \boldsymbol{0}\})$.

The goal of a MOO algorithm is to identify an approximate PF $\mathcal{P}(X_n)$ of the true PF $\mathcal{P}(\mathcal{X})$ within a pre-specified budget of $|X_n| = n$ function evaluations. The quality of a PF is often evaluated using the hypervolume (HV) indicator.

**Definition 2.3.** The *hypervolume indicator*, $\text{HV}(\mathcal{P}(X)|\boldsymbol{r})$ is the $M$-dimensional Lebesgue measure $\lambda_M$ of the region dominated by $\mathcal{P}(X)$ and bounded from below by a reference point $\boldsymbol{r} \in \mathbb{R}^M$.

The reference point is typically provided by the practitioner based on domain knowledge [Yang et al., 2019]. MOO problems are often addressed using evolutionary algorithms (EA) such as NSGA-II [Deb et al., 2002]. However, EAs generally suffer from high sample-complexity, rendering them inapplicable under small evaluation budgets.

#### 2.1.2 Bayesian Optimization

When high sample-efficiency is required, Bayesian optimization (BO) is a popular approach [Frazier, 2018]. BO relies on a probabilistic surrogate model and an acquisition function that uses the surrogate model to provide the utility of evaluating a set of design points on the black-box function. The acquisition function is responsible for balancing exploration and exploitation. In the multi-objective setting, a common approach is to optimize random scalarizations of the objectives [Knowles, 2006, Paria et al., 2020] using a single-objective acquisition function. A more principled approach is to directly optimize the Pareto frontier by selecting candidates with maximum hypervolume improvement either in expectation under the GP posterior [Emmerich et al., 2006] or using Thompson sampling (TS) [Bradford et al., 2018].

### 2.2 RELATED WORK

#### 2.2.1 Multi-objective Bayesian optimization

There have been many recent contributions to multi-objective BO, e.g., Konakovic Lukovic et al. [2020],

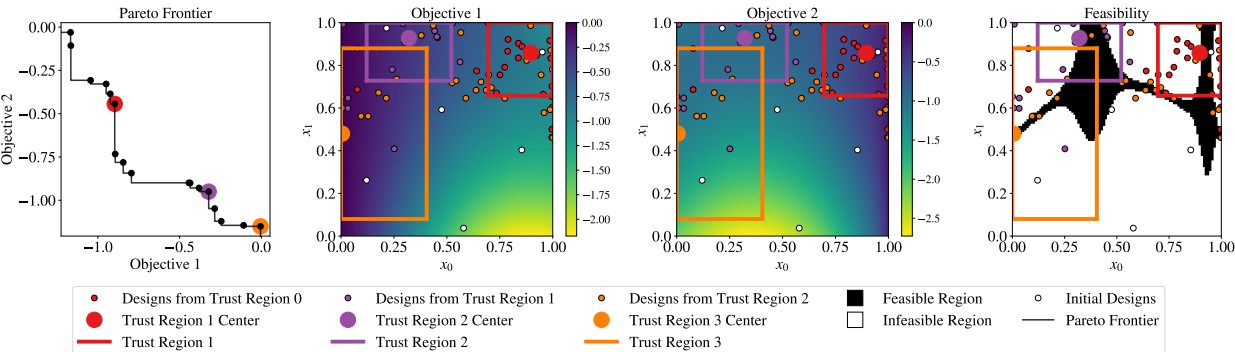

Figure 1: An illustration of MORBO on: 2-objective benchmark problem with 2 parameters and 2 constraints called MW7 [Ma and Wang, 2019] with 3 TRs. The left-most plot illustrates how MORBO's center selection technique centers the TRs at Pareto optimal points across different parts of the Pareto frontier. This encourages MORBO to explore diverse parts of the Pareto Frontier, which is important to identifying the multiple disconnected regions on this MW7 problem. The three right-most plots illustrate the TRs over the design space along with contours of, respectively, the 2 objective metrics and the feasibility metric indicating whether all black-box constraints are satisfied. Note that the TRs overlap with one another and contain data points that were collected by other TRs. Hence, sharing observations collected by different TRs provides local models with more observations than if each local model were only fitted to data collected by its corresponding TR.

Daulton et al. [2020, 2022], Bradford et al. [2018]), but very few methods consider the high-dimensional setting and with large evaluation budgets. All of these methods described below rely on global GP models. As a result, these methods have mostly been evaluated on low-dimensional problems, typically $d \ll 10$ [Konakovic Lukovic et al., 2020, Bradford et al., 2018]. In the multi-objective BO literature, the largest search space we have found consists of 27 parameters [Paria et al., 2020]. Nevertheless, for completeness we review multi-objective BO methods that support generating large batches of designs. DGEMO [Konakovic Lukovic et al., 2020] uses a hypervolume-based objective with heuristics to encourage diversity while exploring the PF.

Parallel expected hypervolume improvement (qEHVI) [Daulton et al., 2020] has strong empirical performance, but its computational complexity scales exponentially with the batch size. qNEHVI [Daulton et al., 2021] improves scalability with respect to the batch size, but like DGEMO and qEHVI, qNEHVI has only been evaluated on low-dimensional search spaces. TSEMO [Bradford et al., 2018] optimizes approximate GP function samples using NSGA-II and uses a hypervolume-based objective for selecting a batch of points from the NSGA-II population. ParEGO [Knowles, 2006] and TS-TCH [Paria et al., 2020] use random Chebyshev scalarizations with parallel expected improvement [Jones et al., 1998] and Thompson sampling—where a design is sampled with probability proportional to a design being optimal [Thompson, 1933]—respectively. ParEGO has been extended to the batch setting in various ways including: (i) MOEA/D-EGO [Zhang et al., 2010], an algorithm that optimizes multiple scalarizations in parallel using MOEA/D [Zhou et al., 2012], and (ii) qParEGO [Daulton et al., 2020], which uses composite

objectives with sequential greedy batch selection under different scalarization weights. Information-theoretic methods, e.g., Hernández-Lobato et al. [2015], Suzuki et al. [2020] have also garnered recent interest.

LaMOO [Zhao et al., 2021] is a recent work that partitions the search space into "good" and "bad" regions and samples new designs from "good" regions using qEHVI or CMA-ES [Hansen, 2007]. However, LaMOO-qEHVI relies on global GPs and is therefore prohibitively time-consuming with large evaluation budgets. In addition, the authors propose to use rejection sampling to enforce that samples are from the, typically non-rectangular, "good" region, but rejection sampling is prohibitively time-consuming in high-dimensional search spaces (see Appendix D.1.1 for further discussion).

### 2.2.2 High-dimensional Bayesian optimization

Two popular approaches for high-dimensional BO are (1) mapping the high-dimensional inputs to a low-dimensional space via a random embedding [Wang et al., 2016, Munteanu et al., 2019, Letham et al., 2020] and (2) exploiting additive structure [Kandasamy et al., 2015, Gardner et al., 2017]. However, both families of methods require strong assumptions on the structure of the problem (low-dimensional linear or additive structure, respectively), and often perform poorly if the assumptions do not hold [Eriksson and Jankowiak, 2021]. This is especially problematic when optimizing multiple objectives since all objectives need to have the same assumed structure, which is unlikely in practice. Eriksson and Jankowiak [2021] leverage a weaker assumption that the objective only depends on a small subset of the parameters and Eriksson et al. [2021] extended this approach to the multi-objective setting, but

this approach requires using computationally-demanding Markov Chain Monte Carlo methods for fitting the model, which is only feasible in the small data regime.

### 2.2.3 Trust Region Bayesian Optimization

Another popular method for high-dimensional BO is TuRBO [Eriksson et al., 2019], which performs BO in local trust regions (TRs) to avoid over-exploration. In contrast with [Zhao et al., 2021] which uses non-rectangular "good" regions, TuRBO uses hyperrectangular TRs, where each TR $\mathcal{T}$ has a center point $x_{center}$ and an edge-length $L \in [L_{min}, L_{max}]$. Each TR maintains success and failure counters that record the number of consecutive samples generated from the TR that improved or failed to improve (respectively) the objective. If the success counter exceeds a predetermined threshold $\tau_{succ}$, the TR length is increased to $\min\{2L, L_{max}\}$ and the counter is reset to zero. Similarly, after $\tau_{fail}$ consecutive failures the TR length is set to $L/2$ and the failure counter is set to zero. Finally, if the length $L$ drops below a minimum edge length $L_{min}$, the TR is terminated and a new TR is initialized.

In contrast with aforementioned methods, TuRBO makes no strong assumptions about the objectives. Although TuRBO has been extended to handle black-box constraints [Eriksson and Poloczek, 2021], to our knowledge, all existing TR-based BO methods target single-objective optimization. In addition, TuRBO does not pass information between TRs, which results in an inefficient use of the evaluation budget; these methods have not observed significant improvement from using multiple TRs. Lastly, even though optimization is restricted to a local TR, TuRBO fits GP models to the entire history of data collected by a single TR which can lead to poor scalability in settings where TRs restart infrequently.

### 2.3 ISSUES WITH SCALARIZED TURBO

Since ParEGO is a well-established method (in low-dimensional settings) that optimizes random Chebyshev scalarizations, a reasonable approach would be to extend TuRBO to the MOO setting by using multiple TRs in parallel where each TR optimizes a different random Chebyshev scalarization of the objectives. However, as we demonstrate in the left subplot of Figure 2, this approach results in a PF with very poor coverage. This is because a single scalarization is used for the lifetime of each TR in order to maintain a stable objective. Optimizing a single scalarization per trust region often leads to better solutions with respect to that scalarization than optimizing the entire PF using a hypervolume-based acquisition functions, which requires exploration of different objective trade-offs. However, if TRs are not restarted frequently (e.g. because TuRBO continues to find better solutions with respect to that scalarization), only a small number of scalarizations will be used, which

can lead to poor coverage of the PF. As shown in Figure 2, we observe that MORBO yields PFs with better coverage (diversity of trade-offs). In addition, the TRs in TuRBO are independent; they do not pass information about evaluated designs and observations, and they do not collaboratively aim to optimize the global PF—rather, they act in isolation to optimize their own objectives. Together, this leads to an inefficient use of the sample budget.

## 3 MORBO

We now introduce MORBO, a *collaborative* multi-TR approach for constrained high-dimensional multi-objective BO. Rather than following TuRBO's approach of employing multiple independent TRs, MORBO shares observations across TRs to provide each TR with all available information about the objectives and constraints relevant for local optimization in the TR. Moreover, MORBO further departs from TuRBO by (1) selecting TR center points in a coordinated fashion to encourage identifying Pareto frontiers with good coverage, (2) choosing new candidate designs by collaboratively optimizing a shared global utility, and (3) employing local models to reduce computational complexity and improve scalability in large data regimes. As shown in the center plot of Figure 2, MORBO identifies a high quality PF with much better coverage than the aforementioned simple TuRBO extension. For the remainder of this section, we describe the core components of MORBO, which are also summarized in Algorithm 1.

### 3.1 COLLABORATIVE BATCH SELECTION VIA GLOBAL UTILITY MAXIMIZATION

Maximizing hypervolume improvement (HVI) has been shown to produce high-quality and diverse PFs [Emmerich et al., 2006]. Given a reference point, the hypervolume improvement from a set of points is the increase in HV when adding these points to the previously selected points. Expected HVI (EHVI) is a popular acquisition function that integrates HVI over the GP posterior. However, maximizing EHVI directly requires re-computing the GP posterior and sampling from it in each gradient step, which becomes prohibitively slow as the number of objectives (and constraints) and in-sample data points increases.

To allow scalability to large batch sizes $q$, we instead use Thompson sampling (TS) to draw $q$ posterior samples from the GP and optimize HVI under each realization. This approach can be viewed as a single-sample approximation of EHVI [Daulton et al., 2021]. We select $q$ points $x_1, ..., x_q$ for the next batch in a *sequential greedy* fashion and condition upon the previously selected points in the batch by computing the HVI with respect to the current PF $\mathcal{P}$. In particular, to select the $i^{th}$ point from a set of $r$ candidate points $\hat{x}_1, \ldots, \hat{x}_r$ we draw a sample from the joint posterior

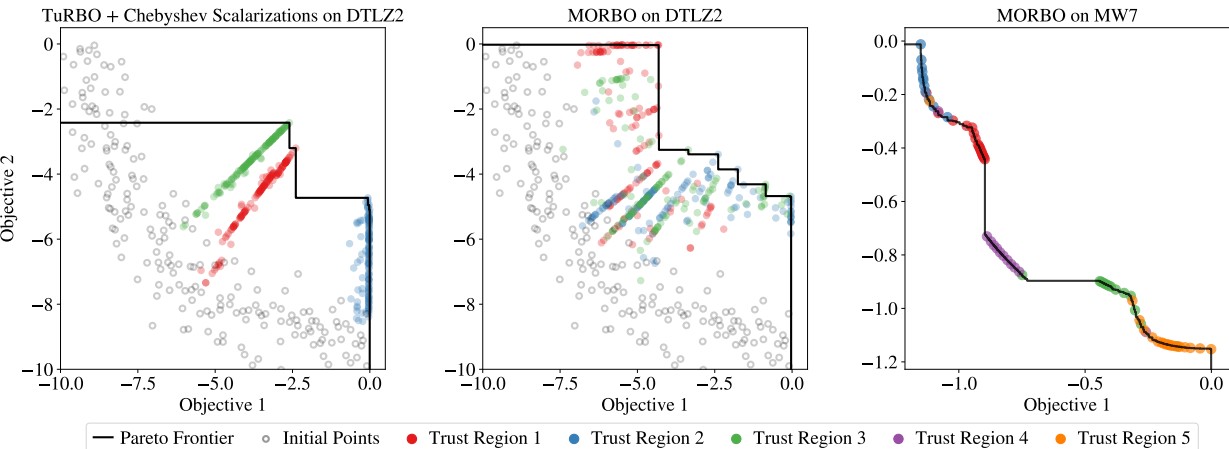

Figure 2: Objective values achieved on a 2-objective DTLZ2 function with $d = 100$ after 600 evaluations, batch size 50, and 3 TRs. The scatter plot illustrates the search behavior. The grey circles indicate the initial space-filling design, which is the same for both methods. The other marker shapes and colors indicate which of the 3 TRs obtained a given solution. The black line indicates the approximate Pareto frontier identified by each method. (Left) A straightforward extension of TuRBO where each TR optimizes a random Chebyshev scalarization of the objectives does not explore the trade-offs between the objectives because the TRs are rarely terminated under this approach, which leads only a few scalarizations being used. (Center) In contrast, MORBO employs a center selection strategy that actively targets under-explored regions of the Pareto frontier and uses a hypervolume-based acquisition function that is known to reward to high quality Pareto frontiers [Zitzler et al., 2003, Couckuyt et al., 2014, Yang et al., 2019] and explores the entirety of the PF. (Right) MORBO can discover disconnected regions of global PF on the MW7 function ($d = 10$, with 2 constraints) by using 5 TRs to locally optimize disjoint regions of PF collaboratively, in parallel. This is stark contrast with TuRBO with Chebyshev scalarizations which the left plot shows yield approximate Pareto frontiers with poor coverage and diversity, even when the true PF is connected and simple.

over $\boldsymbol{f}(\{\boldsymbol{x}_1, \ldots, \boldsymbol{x}_{i-1}\} \cup \{\hat{\boldsymbol{x}}_1, \ldots, \hat{\boldsymbol{x}}_r\})$, which yields the realization $\{\tilde{\boldsymbol{f}}(\boldsymbol{x}_1), \ldots, \tilde{\boldsymbol{f}}(\boldsymbol{x}_{i-1}), \tilde{\boldsymbol{f}}(\hat{\boldsymbol{x}}_1), \ldots, \tilde{\boldsymbol{f}}(\hat{\boldsymbol{x}}_r)\}$. We select the $i^{\text{th}}$ point as the candidate point that maximizes the HVI jointly with the realizations $\tilde{\boldsymbol{f}}(\boldsymbol{x}_1), \ldots, \tilde{\boldsymbol{f}}(\boldsymbol{x}_{i-1})$ of the previously selected points as shown in Figure 1. Conditioning on the previously selected points and computing the HVI under a sample from the joint posterior over the previously selected points and the discrete set of candidates leads to more diverse batch selection compared to selecting each point independently. Moreover, this approach effectively lets TRs collaboratively maximize the global HVI utility function. Using this global utility, an individual TR considers the iteration a success if at least one proposed candidate improves the global HV and a failure otherwise.

Another benefit of HV-based acquisition functions is that they naturally provide utility values for set of points, which enables the TRs to target different parts of the PF. This is particularly appealing in settings where the PF may be disjoint or may require exploring different parts of the search space. As shown in the right plot of Figure 2, MORBO recovers diverse regions of a disconnected PF. Lastly, we note that this batch selection strategy also allows to straightforwardly implement *fully asynchronous* optimization, where evaluations are dispatched to different "workers" and new candidates are generated whenever there is capacity in the

worker pool. In the asynchronous setting, success/failure counters and TRs can be updated after every $q$ observations are received, and intermediate observations can immediately be used to update the local models.

## 3.2 COORDINATED TRUST REGION CENTER SELECTION

In (constrained) single-objective optimization, previous work centers the local TR at the best (feasible) observed point. However, in the multi-objective setting, there is typically no single best solution. Assuming noise-free observations, MORBO selects the center to be the feasible point on the PF with maximum hypervolume contribution (HVC) [Beume et al., 2007, Loshchilov et al., 2011]. If there is no feasible point, MORBO chooses the point with the smallest total constraint violation (see Appendix B for details on center selection with constraints). Given a reference point, the HVC of a point on the PF is the reduction in HV if that point were to be removed; that is, the HVC of a point is its exclusive contribution to the PF. Centering a TR at the point with maximal HVC collected by that TR promotes coverage across the PF, as points in crowded regions will have lower contribution. MORBO selects TR centers based on their HVCs in a sequential greedy fashion, exclud-

**Algorithm 1:** Summary of MORBO

---

**Input:** Objective functions $f$, Number of trust region $n_{\text{TR}}$, Initial trust region length $L_{\text{init}}$, Maximum trust region length $L_{\text{max}}$, Minimum trust region length $L_{\text{min}}$.

**Output:** Approximate Pareto frontier $\mathcal{P}_n$

1   Evaluate an initial set of points and initialize the trust regions $\mathcal{T}_1, ..., \mathcal{T}_{n_{\text{TR}}}$ using the center selection procedure described in Section 3.2 and mark center points as unavailable for other trust regions.

2   $X_0 \leftarrow \emptyset, Y_0 = \emptyset, t \leftarrow 1$

3   **while** *budget not exhausted* **do**

4      Fit a local model within each trust region.

5      Select $q$ candidates using the sequential greedy HVI procedure described in Section 3.1.

6      Evaluate candidates on the true objective functions and obtain new observations.

7      **for** $j = 1, ..., n_{TR}$ **do**

8         Update trust regions with new observations as described in Section 3.

9         Increment success/failure counters as described in Section 3 for observations from $T_j$.

10        Update edgelength $L_j$ for $\mathcal{T}_j$.

11        **if** $L_j < L_{min}$ **then**

12           Terminate $\mathcal{T}_j$.

13           Fit GP to restart points $\mathcal{D}_{t-1} = (X_{t-1}, Y_{t-1})$: $\boldsymbol{f}_{t-1} \sim P(\boldsymbol{f}|\mathcal{D}_{t-1})$.

14           Sample $\boldsymbol{\lambda} \sim S_+^{M-1}$ and $\tilde{\boldsymbol{f}}_{t-1} \sim P(\boldsymbol{f}|\mathcal{D}_{t-1})$, where $S_+^{M-1} = \{\boldsymbol{w} \in \mathbb{R}_+^M : ||\boldsymbol{w}||_2 = 1\}$.

15           $\boldsymbol{x}_t \leftarrow \arg\max_{\boldsymbol{x} \in \mathcal{X}} s_{\boldsymbol{\lambda}}[\tilde{\boldsymbol{f}}_{t-1}(\boldsymbol{x})]$, where $s_{\boldsymbol{\lambda}}[\boldsymbol{y}] = \min_m(\max(\frac{y_m}{\lambda_m}, 0))^M$ and $\cdot_i$ denotes the $i^{\text{th}}$ element.

16           Evaluate $\boldsymbol{x}_t$ on the true objective functions and obtain new observation $\boldsymbol{y}_t$.

17           Reinitialize $\mathcal{T}_j$ with edgelength $L_{\text{init}}$ centered at the $\boldsymbol{x}_t$.

18           Set $X_t \leftarrow X_{t-1} \cup \{\boldsymbol{x}_t\}, Y_t \leftarrow Y_{t-1} \cup \{\boldsymbol{y}_t\}, t \leftarrow t+1$.

19        Update center to the available point with maximum HVC (globally if $\mathcal{T}_j$ was terminated otherwise within $\mathcal{T}_j$).

20   **return** *Approximate PF across observed function values.*

---

ing points that have already been selected as the center for another TR.

### 3.3   LOCAL MODELING

Most BO methods use a single global GP model, often with a stationary kernel (e.g. Matérn-$5/2$) using automatic relevance determination (ARD) fitted to all observations collected so far. While a global model is necessary for most BO methods, MORBO only requires each model to be accurate within the corresponding TR. To increase scalability, we employ local modeling where we only include the observations contained within a local modeling hypercube with edge length $2L$. The motivation for using the observations from a slightly larger hypercube is to improve the model close to the TR boundary.

In previous trust region BO works [Eriksson et al., 2019, Eriksson and Poloczek, 2021, Wan et al., 2021], each TR uses a GP that is fitted to the all observations collected by that TR (rather than only a set of local observations in or near the TR), which leads to scalability issues due to the cubic time complexity of GP inference if the TR collects many observations. In addition, fitting a GP solely to data collected by a single TR ignores observations collected by other TRs and makes inefficient use of the sampling

budget. In contrast, MORBO shares observations across TRs and employs local models, where models are fit to all observations within a hypercube with edge length $2L$. This significantly reduces the computational cost since exact GP fitting scales cubically with the number of data points. Under limited assumptions on the distribution of data across TRs, using local models results in speedups of $O(n_{\text{TR}}^2/\eta^3)$, where $\eta$ is the average number of TR modeling spaces a data point resides in. Empirically, we demonstrate (see Figure 3 in Appendix F) that $\eta < 1$ as the optimization progresses and the TRs shrink, and we find that this translates into speedups of two orders of magnitude relative to global modeling as shown in Appendix F.2. See Appendix E for more details on the complexity.

### 3.4   RE-INITIALIZATION STRATEGY

Although MORBO performs local optimization within a TR, we ensure global optimization by re-initializing TRs using a principled technique based on hypervolume scalarizations [Zhang and Golovin, 2020]. A HV scalarization is defined as $s_{\boldsymbol{\lambda}}[\boldsymbol{y}] = \min_m(\max(\frac{y_m}{\lambda_m}, 0))^M$, where $\cdot_m$ denotes the $m^{\text{th}}$ component [Zhang and Golovin, 2020]. Let $\mathcal{D}_{t-1} = (X_{t-1}, Y_{t-1})$ be the set of previous re-initialization (restart) points $X_{t-1} = \{\boldsymbol{x}_i\}_{i=1}^{t-1}$ and corresponding observations $Y_{t-1} = \boldsymbol{f}(X_{t-1})$, where $X_0 = \emptyset$ and $Y_0 = \emptyset$.

Given $\mathcal{D}_{t-1}$, we determine the center point $x_t$ of the new TR by maximizing a random HV scalarization of the objectives under a posterior sample from a global GP posterior conditioned on $D_{t-1}$: $\tilde{\boldsymbol{f}} \sim P(\boldsymbol{f}|\mathcal{D}_{t-1})$. This ensures that TRs are initialized in diverse parts of the objective space and yields a global optimization performance guarantee (Section 4).

## 4 THEORETICAL ANALYSIS

We analyze the performance of MORBO in terms of its cumulative HV regret. The instantaneous HV regret $R(\mathcal{P}_t)$ after $t$ TR restarts is defined as the difference in HV dominated by the true Pareto frontier $\mathcal{P}^*$ and the approximate Pareto frontier $\mathcal{P}_t$: $R(\mathcal{P}_t) = \text{HV}(\mathcal{P}^*) - \text{HV}(\mathcal{P}_t)$. The (cumulative) HV regret after $T$ restarts is the sum of the instantaneous regret over all restarts: $R_T = \sum_{t=1}^{T} R(\mathcal{P}_t)$. First, we show that a TR will only evaluate a finite number of samples before restarting.

**Lemma 4.1.** *Let $\boldsymbol{f} \in [0, B]^M$, and assume that MORBO only considers a newly evaluated sample to be an improvement (for updating the corresponding TR's success and failure counters) if it increases the HV by at least $\delta \in \mathbb{R}^+$ and assume that success counter threshold $\tau_{succ} = \infty$.[1] Then each TR will only evaluate a finite number of samples.*

The proof is given in Appendix C. Having established that TRs only evaluate a finite number of designs, we now bound the hypervolume regret with respect to the number of restarted TRs. The bound leverages the kernel-dependent maximum information gain $\gamma_T$—which measures the decrease in uncertainty after $T$ observations —and is commonly used to analyze regret in BO [Srinivas et al., 2010].

**Theorem 4.1.** *Let $\boldsymbol{f} \in [0, B]^M$ for $B > 0$ and let each component $f^{(m)}$ for $m = 1, ..., M$ follow a Gaussian distribution with marginal variances $\sigma \leq 1$ and independent observation noise $\epsilon_m \sim \mathcal{N}(0, \sigma_m^2)$ such that $\sigma_m^2 \leq \sigma^2 \leq 1$. Let $\mathcal{P}_t$ denote the Pareto frontier over $\boldsymbol{f}(X_t)$, where $X_t$ is the set of TR re-initialization points after $t$ TRs have been restarted. Suppose further that the conditions of Lemma 4.1 hold. Then, the cumulative hypervolume regret $R_T$ of MORBO after $T$ restarts is bounded by:*

$$R_T \leq M^2(\sqrt{2e\pi}B/2)^M \sqrt{d\gamma_T T \ln(T)}.$$

Up to logarithmic terms, this regret bound is on the order of $\tilde{\mathcal{O}}(\sqrt{T})$. This bound is significant because, to our knowledge, Zhang and Golovin [2020] is the only other work to bound the HV regret of multi-objective BO algorithms. This makes MORBO the first sample-efficient large-scale, MOO algorithm with bounded regret. The proof, given in

---

[1] As stated in Appendix D, we use $\tau_{\text{succ}} = \infty$ in all of our experiments.

Appendix C, leverages the hypervolume regret bound from Zhang and Golovin [2020]. However, our regret bound is with respect to the number of restart points (rather than evaluations)—a difference that can be viewed as a cost of focusing on large-scale problems which BO with global GPs cannot address. Moreover, our regret analysis in terms of the number of restarts is similar to the convergence guarantees of gradient-based TR optimization methods [Yuan, 1999] and can be viewed as a multi-objective analogue of the performance guarantees of recent single-objective BO-based TR methods [Wan et al., 2021].

## 5 EXPERIMENTS

We evaluate MORBO on an extensive suite of benchmarks with various numbers of input parameters ($d$), objectives ($M$), and constraints ($V$). In Appendix F.1, we consider a vehicle ($d = 5$) and a welded beam ($d = 4$, $V = 4$) design problem to show that MORBO is competitive with other algorithms on problems it was not designed for. We consider three challenging real-world problems: a trajectory planning problem ($d = 60$), a problem of designing optical systems for AR/VR applications ($d = 146$), and an automotive design problem ($d = 222$, $V = 54$) . In addition, we evaluate MORBO on DTLZ3, DTLZ5, and DTLZ7 problems with 2/4 objectives (6 problems in total) in Appendix F.

We compare MORBO to multi-objective BO methods ($q$NEHVI, $q$ParEGO, TS-TCH, TSEMO, DGEMO, MOEA/D-EGO), recent work leveraging search space partitioning (LaMOO-CMAES, LaMOO-$q$NEHVI), a widely used evolutionary algorithm (NSGA-II), and Sobol—a quasi-random baseline where designs are sampled from a scrambled Sobol sequence [Owen, 2003] (see Appendix D for more details on the methods). MORBO is implemented using BoTorch [Balandat et al., 2020] and the code will be made publicly available soon. We run all methods for 20 replications and initialize them using the same quasi-random initial points for each replication. We use the same hyperparameters for MORBO on all problems and conduct analyze the sensitivity of MORBO to its hyperparameters in Figure 4. See Appendix D for details on the experiment setup. All experiments used a Tesla V100 SXM2 GPU (16GB RAM).

### 5.1 LARGE-SCALE REAL-WORLD PROBLEMS

**Trajectory Planning** We consider a trajectory planning problem similar to the rover trajectory planning problem considered in [Wang et al., 2018]. As in the original problem, the goal is to find a trajectory that maximizes the reward when integrated over the domain. The trajectory is determined by fitting a B-spline to 30 design points in the 2-objective plane, which yields a 60-dimensional optimization problem. In this experiment, we constrain the trajectory to begin at the pre-specified starting location, but we do not

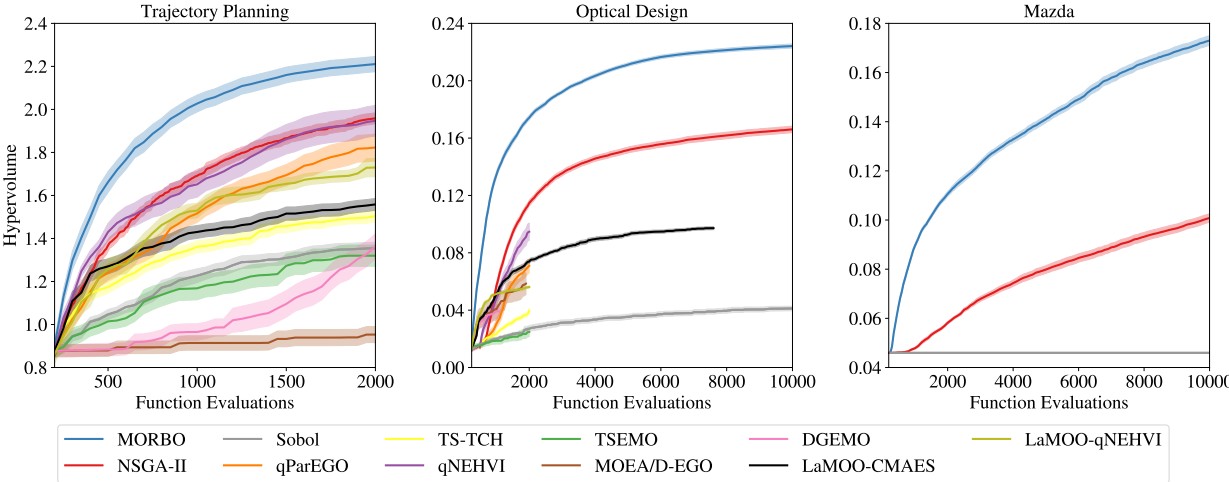

Figure 3: (Left) MORBO outperforms other methods on the trajectory planning problem ($d = 60$). (Middle) Illustration of the results on the Optical design problem ($d = 146$). NSGA-II performs better than the BO baselines but is not competitive with MORBO. (Right) MORBO shows compelling performance on the Mazda vehicle design problem ($d = 222$) with $54$ black-box constraints. For all plots, we show the mean and one standard error of the mean over 20 replications.

require it to end at the desired target location. In addition to maximize the reward of the trajectory, we also minimize the distance from the end of the trajectory to the intended target location. Intuitively, these two objectives are expected to be competing because reaching the exact end location may require passing through areas with lower associated reward. The results from 2,000 evaluations using batch size $q = 50$ and 200 initial points are presented in Figure 3, which shows that MORBO performs the best and even state-of-the-art methods such as $q$NEHVI do not out perform NSGA-II.

**Optical design problem**    We consider the problem of designing an optical system for an augmented reality (AR) see-through display. This optimization task has 146 parameters describing the geometry and surface morphology of multiple optical elements in the display stack. Several objectives are of interest in this problem, including display efficiency and display quality. Each evaluation of these metrics requires a computationally intensive physics simulation that takes several hours to run. In this benchmark, the task is to explore the Pareto frontier between display efficiency and display quality (both objectives are normalized w.r.t. the reference point). We consider 250 initial points, batch size $q = 50$, and a total of 10,000 evaluations. This is out of reach for the other BO baselines due to runtime considerations, and so we run $q$NEHVI, $q$ParEGO, TS-TCH, TSEMO, MOEA/D-EGO, for 2,000 evaluations and DGEMO for 1,000 evaluations. We were only able to run LaMOO-CMAES for $7,600$ evaluations before it overflowed GPU memory. Figure 3 shows that MORBO achieves substantial improvements in sample efficiency compared to NSGA-II. Furthermore, observe that no other baselines are competitive with NSGA-II except in the very small sample regime (less than $500$ evaluations).

**Mazda vehicle design problem**    We consider the 3-car Mazda benchmark problem [Kohira et al., 2018]. This challenging MOO problem involves tuning 222 decision variables that represent the thickness of different structural parts. The goal is to minimize the total vehicle mass of the three vehicles (Mazda CX-5, Mazda 6, and Mazda 3) as well as maximizing the number of parts shared across vehicles. Additionally, there are 54 black-box output constraints (evaluated jointly with the two objectives) that enforce that designs meet performance requirements such as collision safety standards. This problem is, to the best our knowledge, the largest MOO problem considered by any BO method and requires fitting 56 GP models to the objectives and constraints. The original problem underlying the Mazda benchmark was solved on what at the time was the world's fastest supercomputer and took around 3,000 CPU years to compute [Oyama et al., 2017]. We consider a budget of $10,000$ evaluations using batches of size $q = 50$ and 300 initial points.

Figure 3 demonstrates that MORBO clearly outperforms the other methods. A feasible design satisfying the black-box constraints was provided to all methods for all replications as part of the initial 300 design points. However, in subsequent evaluations Sobol did not find another feasible design, illustrating the challenge of satisfying the 54 constraints. While NSGA-II made progress from the initial feasible solution, it is not competitive with MORBO. NSGA-II and Sobol are the only applicable baselines because standard multi-objective BO methods are impractically slow with 56 *global* GPs and LaMOO does not support black-box constraints.

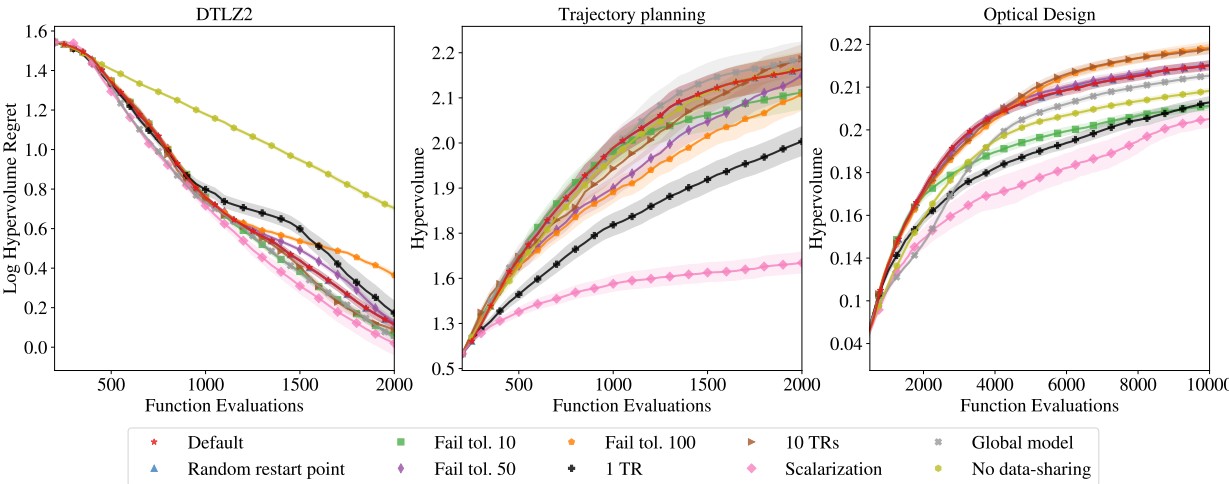

Figure 4: We investigate the sensitivity of MORBO with respect to its hyperparameters. We observe that using multiple TRs performs significantly better than using a single TR and that data-sharing and the use of a hypervolume based acquisition function are important components of MORBO.

## 5.2 ABLATION STUDY

Finally, we study the sensitivity of MORBO with respect to the number of TRs ($n_{\text{TR}}$), the failure tolerance ($\tau_{\text{fail}}$), and sharing observations across TRs, local modeling, HVI acquisition function, and the re-initialization strategy. Using several TRs allows MORBO to explore different parts of the search space that potentially contribute to different parts of the Pareto frontier. The failure tolerance controls how quickly each TR shrinks: A large $\tau_{\text{fail}}$ leads to slow shrinkage and potentially too much exploration, while a small $\tau_{\text{fail}}$ may cause each TR to shrink too quickly and not collect enough data. MORBO uses 5 TRs and $\tau_{\text{fail}} = \max(10, \frac{d}{3})$ by default, similar to what is used by Eriksson et al. [2019].

We consider the DTLZ2 problem ($d = 100$, $M = 2$), the trajectory planning problem ($d = 60$, $M = 2$), and the optical design problem ($d = 146$, $M = 2$). Figure 4 shows that MORBO with the default settings performs well on all three problems. We observe that multiple TRs and the HVI acquisition function are important as neither a single TR nor a Chebyshev scalarization performs well. The performance of MORBO is robust to the choice of failure tolerance except for on the optical design problem where using a value of 10 is clearly worse than the default and causes the TRs to shrink too quickly. Not sharing data between TRs results in inferior results on the DTLZ2 and optical design problems. While using a global GP model achieves good results on the DTLZ2 and trajectory planning problems, it does not perform as well on the optical design problem. A global GP also comes at a high computational cost. Using a global GP, running MORBO with a budget of 10,000 evaluations on the optical design problem required 30 hours of computational overhead, whereas MORBO did 10,000 evaluations in less than an hour using local models. Lastly, we find consistently strong performance for both our default HV scalarization-based re-initialization strategy and a strategy that selects a new design at random (denoted as "Random restart points"). The former allows us to bound MORBO's regret.

## 6 DISCUSSION

We proposed MORBO, an algorithm for multi-objective BO over high-dimensional search spaces. By using a coordinated, collaborative multi-trust-region approach with scalable local modeling, MORBO scales gracefully to high-dimensional problems and high-throughput settings. In a comprehensive experimental evaluation, we showed that MORBO allows us to *effectively tackle important real-world problems that were previously out of reach for existing BO methods*. We showed that MORBO achieves substantial improvements in sample efficiency compared to existing state-of-the-art methods such as evolutionary algorithms. Due to the lack of alternatives, NSGA-II has been the method of choice for many practitioners, and we expect MORBO to provide practitioners with significant savings in terms of time and resources across the many disciplines that require solving challenging optimization problems.

However, there are some limitations to our method. Although MORBO can handle a large number of black-box constraints, using hypervolume-based acquisition means the computational complexity scales poorly with the number of objectives. Furthermore, MORBO is optimized for the large-batch high-throughput setting and other methods may be more suitable for and achieve better performance on low-dimensional problems with small evaluation budgets.

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
