# OpenReview forum: "Multi-Objective Bayesian Optimization over High-Dimensional Search Spaces"
_auai.org/UAI/2022/Conference — UAI 2022 Oral_

### Official Review · Reviewer_cMEA · 2022-04-11

**Q2(1) Originality/Novelty:** 3
**Q2(2) Significance/Impact:** 3
**Q2(3) Correctness/Technical Quality:** 3
**Q2(6) Clarity Of Writing:** 2
**Q6 Overall Score:** 7
**Q8 Confidence In Your Score:** 3

**Q1 Summary And Contributions:**

The authors consider multi-objective optimization over high-dimensional parameter spaces, specifically when evaluations are costly. Good sample efficiency makes Bayesian optimization (BO) a promising candidate for solving such problems, but existing BO techniques don’t scale to high dimensions – forcing the use of (sample-inefficient) evolutionary techniques instead. The authors present a new BO algorithm, the first to perform well in the high-dimensional case.

**Q10 Ethical Concerns (Optional):**

No.

**Q2 Assessment Of The Paper:**

More detailed information regarding each of these aspects is given below:

**Q2(4) Quality Of Experiments (Optional):**

3: Good: The experimental evaluation is adequate, and the results convincingly support the main claims.

**Q2(5) Reproducibility:**

3: Good: Key resources (e.g., proofs, code, data) are available and key details (e.g., proofs, experimental setup) are sufficiently well-described for competent researchers to confidently reproduce the main results.

**Q3 Main Strengths:**

Novelty:
As far as I can assess, this is indeed the first algorithm to fill the niche described (multi objective optimization at high evaluation cost and high-dimensionality).

Impact:
That means it could potentially have a good impact if it indeed works as well as described, at least within the niche of problems identified by the authors.

Experiments:
The authors compare their method with a large set of other BO methods and also with an evolutionary algorithm. They also include a sensitivity analysis to show which hyperparameters are important, which is valuable to readers who might wish to use this method.

Reproducibility:
The authors state that the code will be available as open-source software. I obviously cannot assess this now, but if the authors indeed share the code the work should be reproducible.


**Q4 Main Weakness:**

Impact cannot entirely be assessed because it is not clear how often such problems (multi-objective, high-dimensional, costly evaluation) occur. This would be good to contextualize further in the paper; see also detailed comments.

My biggest issue with the paper was the clarity of writing. While the paper is overall well-written, several key concepts were unclear to me (or only after several reads), and some terms were used before they were introduced. See Q5 for details; there I have tried to explain where I was confused, asked for some clarifications, and offered suggestions on how to improve this.


**Q5 Detailed Comments To The Authors:**

Major points/Qs:
1.	To clarify the potential impact of the work, it would be good to expand more on the problems that this method can potentially solve. Specifically, to help readers assess impact it would be good to discuss:

a.	How common/rare are these “high-dimensional, expensive eval” problems that MORBO is designed for?

b.	MORBO works well on the high-dimensional problems tested here. But how “high-dimensional” can you go – is this dimensionality approaching the max of what is currently studied in literature (not just for BO, but also eg NSGA-II)?

c.	The paper mentions a few times that it is assumed that objective evaluations are noiseless. To what extent does MORBO generalize to cases with noise? Or to what extent are typical real-world problems noiseless?

2.	While reading the paper I was confused which aspects of the algorithm are local and which are global. E.g. on page 4 (but also in other places): “…(2) choosing new candidate designs by collaboratively optimizing a shared global utility, and (3) employing local models to reduce computational complexity and improve scalability in large data regimes.” Initially, I was confused because these two statements seemed to be at odds. But since I am not a BO expert, I’d first like to check if I am correct in the following:

a.	BO selects new evaluations by predicting *beforehand* what the “added value” (HVI) of a new point will be, in order to run expensive evaluations only on the most promising candidates. (Is this correct Y/N?)

b.	The question is then how to get these predictions of the objectives without actually evaluating. BO typically uses GP prediction models fitted on earlier observations, but this is slow. To speed up this process, MORBO doesn’t fit one GP model but several local ones that each use only nearby datapoints. This is the “local” aspect of the algorithm, but there is some “global” sharing in the sense that TRs can also use datapoints from nearby/overlapping TRs for fitting their “local” GP. (Is this correct Y/N?)

c.	Once fit, these GP models predict function evaluations of new candidate solutions, which in turn provides their predicted “utility” (HVI). The HVI is global because it considers the entire pareto frontier discovered so far. (Is this correct Y/N, or if N, is the HVI also computed only within each TR? If it’s the latter, to what extent does that make a difference?)

d.	Given the predicted utility/HVI, MORBO selects the most promising candidates to *actually* run the (expensive) evaluation on (confirming the predictions). But how are these “most promising candidates” selected – i.e.:

i.	Do you globally pick the q most promising candidates regardless of which TR they are in?
ii.	Or do you pick the best ones locally within each TR? And then the “global” aspect is that you select the “globally best” candidate whenever a new TR is initiated?

e.	Even if my understanding in a-d is correct, it took me a few reads to get to this understanding. I would advise to state the “local” and “global” aspects more explicitly in the paper, in one central spot (now, the explanations are in different parts of the paper which makes it harder to understand). In particular, the authors might wish to make explicit the difference between predicting the *function evaluations* (point b, through local GPs), assessing the *utility* (HVI, point c, which I assume is a global measure), and using the utility to predict *best candidates* (within/across TRs?).

f.	Related comment: on p5 “While each TR … only uses observations collected by that TR, each TR still uses a global GP fitted to the entire history of that TR”. This was confusing to me because of the word “global”. I guess what you mean here is that TuRBO uses all observations previously used by the TR, while MORBO discards observations as TRs shrink – is this correct Y/N? If Y, TuRBO still uses only observations from the TR, so is not really “global” in that sense. I would suggest using a different word here as “global” is confusing (unless I have misunderstood).

3.	The authors state that sharing observations between closeby/overlapping TRs is important for their algorithm (if I understand correctly, they disable this in the “no data sharing” condition in Fig 4 and find a performance decrease in some problems). However, this implies that points frequently are used by the GPs of multiple TRs, which at first glance seems contradictory to the idea of fitting local GPs in the first place (in the most extreme case, all points would be included in all TRs, which would be equivalent to using a global GP). This paradox is resolved in the appendix, where the authors show that the eta metric shrinks during the algorithm.

a.	Does this mean that data sharing is more important during the initial part of the algorithm, while the “time-bonus” mostly occurs later when eta decreases (and there will be very little sharing of points between TRs) – Y/N?

b.	Indeed this is what it would seem from Fig 10, where eta starts close to 4 (very close to the total #TRs, 5, so essentially the algorithm does start with a global GP, right?)

c.	If so, this might again be worth stating more explicitly in the main text to avoid confusion.

4.	In Fig 2 it is shown that MORBO explores the pareto frontier more thoroughly, but it also seems to come to a different frontier. E.g. the red/green lines in the left panel seem to reach further than the MORBO PF. Does this mean that MORBO trades off the increased exploration for decreased exploitation, Y/N? If so, this might be something to mention more explicitly in the discussion.

5.	In the experiments, it is shown that MORBO outperforms/can compete with existing methods for several problems. In the sensitivity analysis they show how MORBO depends on its hyperparameters – but what about the hyperparameters of the other methods? In the appendix it is stated that these were often chosen as the default values. But to what extent are the authors confident that these default values are suitable for the problems tested here, and that these methods might not perform better with different hyperparameters? With this I don’t mean that the authors should do an extensive hyperparameter screening, but I would like to see some discussion of why they think the choices here are reasonable.

6.	It is not entirely clear to me how the described Thompson sampling works. For example:

a.	Minor remark: it is used in section 2 but only explained in section 3; this might be good to reorder (or to refer to the section where it is explained).

b.	It is not entirely clear to me how it differs from EHVI. The authors state that EHVI “requires re-computing the GP posterior and sampling from it in each gradient step”, but also that TS is used to “optimize HVI under each realization”. What is the difference? So how does TS sampling solve the problem?

c.	What does “conditioning on the previously selected points” entail? Is this just that you compare the added value of a new candidate point on top of the previous points?

d.	Fig 5 – “To select the 4th point, the HVI of each candidate is evaluated jointly with the red, blue, orange, and green points”. But if you select greedily, wouldn’t you first combine blue/orange/green with red individually? I.e. you assess whether blue + red or orange + red or green + red is better, right? Or did I misunderstand?

7.	Appendix E2: why are all wall times for NSGA-II zero in table 5? And if possible I think it would be good to include NSGA-II in table 4.

Minor remarks on presentation (no response needed):

-	The paper contains quite many acronyms and terms. While I understand that this saves some space, it also makes the paper somewhat inaccessible for readers who are not already familiar with the acronyms used. The authors might wish to reevaluate their list of acronyms, focusing on the ones that are crucial/used very often and writing out other terms in full.

-	Figure 1 provides a nice overview of the method, but is presented early on in the paper before crucial terms like “pareto frontier” and “trust region” are actually explained. To ensure readability for a broader audience, it might be worth to provide a simpler, intuitive explanation here, or to move this figure to a later place when the relevant terms have been explained.

-	The authors mention that the hypervolume reference point is “provided by the practitioner based on domain knowledge” – if there is space, might be worth expanding a little on this (chosen how? How much does this choice affect results?)

-	Small note on notations like “the left plot of Figure 2”, “the center plot of Figure 2”, .. etc -> it might be easier to just label them 2A, 2B, 2C etc.


**Q7 Justification For Your Score:**

Impact: as far as I can see, this method indeed fills a niche for which existing methods are lacking. But as I don’t know how common such tasks are (see Q4,Q5), I cannot assess impact.

The experimental evaluation is also extensive.

The clarity of writing made the paper hard to read for me as a non-expert, but I think that for scientists within the field this will be less of a problem.


**Q9 Complying With Reviewing Instructions:**

1: Yes.

---

### Official Review · Reviewer_KWKJ · 2022-04-17

**Q2(1) Originality/Novelty:** 3
**Q2(2) Significance/Impact:** 3
**Q2(3) Correctness/Technical Quality:** 3
**Q2(6) Clarity Of Writing:** 3
**Q6 Overall Score:** 6
**Q8 Confidence In Your Score:** 4

**Q1 Summary And Contributions:**

This paper solves a problem of multi-objective Bayesian optimization over high-dimensional search spaces, by considering multiple local regions of the design space using a coordinated strategy. The authors show the theoretical results regarding cumulative regret bounds and the numerical results on high-dimensional spaces and a large number of query points.

**Q2 Assessment Of The Paper:**

More detailed information regarding each of these aspects is given below:

**Q2(4) Quality Of Experiments (Optional):**

4: Excellent: The experimental evaluation is comprehensive and the results are compelling.

**Q2(5) Reproducibility:**

3: Good: Key resources (e.g., proofs, code, data) are available and key details (e.g., proofs, experimental setup) are sufficiently well-described for competent researchers to confidently reproduce the main results.

**Q3 Main Strengths:**

+ It solves an interesting problem.
+ The paper is generally well-written.
+ It successfully solves a problem of high-dimensional search spaces, which is capable of evaluating a large number of query points, in practice.

**Q4 Main Weakness:**

- The proposed method is somewhat incremental, although it does not degrade the contributions much.

**Q5 Detailed Comments To The Authors:**

This paper is generally well-written and the contributions of this work is crystal clear, regardless of the amount of contributions. Since the authors demonstrate that the proposed method successfully optimizes the problem defined on high-dimensional search spaces and the evaluation budget is huge, which implies that a large number of query points exist, I think this work is a good paper.

And, I am somewhat confused, for Figure 1, Feasible Region (black) and Infeasible Region (white) are correct? If they are correct, what is the meaning of both? How can we evaluate a query point in the infeasible region?

**Q7 Justification For Your Score:**

As described above, since the proposed method successfully solves the problems of interest, I would like to recommend borderline acceptance for this paper.

**Q9 Complying With Reviewing Instructions:**

1: Yes.

---

### Official Review · Reviewer_4vL1 · 2022-04-18

**Q2(1) Originality/Novelty:** 3
**Q2(2) Significance/Impact:** 3
**Q2(3) Correctness/Technical Quality:** 3
**Q2(6) Clarity Of Writing:** 3
**Q6 Overall Score:** 7
**Q8 Confidence In Your Score:** 2

**Q1 Summary And Contributions:**

The author(s) proposed MORBO, a variant of the BO algorithm that can scale to multiple objective functions and many parameters. The proposed method is backed both theoretically and empirically.

**Q2 Assessment Of The Paper:**

More detailed information regarding each of these aspects is given below:

**Q2(4) Quality Of Experiments (Optional):**

3: Good: The experimental evaluation is adequate, and the results convincingly support the main claims.

**Q2(5) Reproducibility:**

3: Good: Key resources (e.g., proofs, code, data) are available and key details (e.g., proofs, experimental setup) are sufficiently well-described for competent researchers to confidently reproduce the main results.

**Q3 Main Strengths:**

- The paper is well-written. I can get the main idea of the paper even though I am not very familiar with the literature.
- Extensive experiments are conducted and the result looks promising.


**Q4 Main Weakness:**

- Accuracy of the statement.


**Q5 Detailed Comments To The Authors:**

The author(s) emphasized that MORBO is sample efficient and works in the high-dimensional setting. It seems to me that Theorem 4.1 is not strong enough for such a statement since the regret exponentially depends on the dimension size. A more accurate would be: (i) the author(s) give a regret bound for MORBO, which is non-trivial for the analysis of multi-objective BO; (ii) extensive experiments suggest MORBO is more sample-efficient than other baseline methods.

Minor:
- The algorithm should be placed in the main context.

**Q7 Justification For Your Score:**

The paper is well-written and the experimental result looks promising. The author(s) also claim that they will make the code an open-source software. Overall, I think this paper is a decent contribution to the field.

**Q9 Complying With Reviewing Instructions:**

1: Yes.

---

### Official Review · Reviewer_JhWM · 2022-04-18

**Q2(1) Originality/Novelty:** 3
**Q2(2) Significance/Impact:** 4
**Q2(3) Correctness/Technical Quality:** 3
**Q2(6) Clarity Of Writing:** 2
**Q6 Overall Score:** 8
**Q8 Confidence In Your Score:** 2

**Q1 Summary And Contributions:**

Make Bayesian Optimization more applicable to **high dimensional**, multi-objective, constrained problems, tackling previously-out-of-reach problems. The proposed algorithm (MORBO) improves existing BO work by
- sharing observations across Trusted Regions (TR)
- selecting TR centers in a coordinated fashion for better coverage
- choosing new candidate designs by optimizing a shared utility
- employing local models to improve scalability

They show theoretical bounds and strong empirical results

**Q2 Assessment Of The Paper:**

More detailed information regarding each of these aspects is given below:

**Q2(4) Quality Of Experiments (Optional):**

3: Good: The experimental evaluation is adequate, and the results convincingly support the main claims.

**Q2(5) Reproducibility:**

3: Good: Key resources (e.g., proofs, code, data) are available and key details (e.g., proofs, experimental setup) are sufficiently well-described for competent researchers to confidently reproduce the main results.

**Q3 Main Strengths:**

Extensive empirical comparison to other work (experiments) and informative related work summary.

Strong results, on convincing problems. An advancement of the current state-of-the-art, and since the authors stated that they will release the code upon publication, I believe this work will likely have an impact.

**Q4 Main Weakness:**

The algorithmic description is quite high level. For readers unfamiliar with Bayesian Optimisation, only reading this paper is insufficient to completely understand in detail the procedure proposed in this paper. Pseudocode of the algorithm (Algorithm 1 included in appendix) helped to somewhat improve this, but still..

**Q5 Detailed Comments To The Authors:**

I was added later as an emergency reviewer and must admit that I am less familiar with the field, and did not have much time to further investigate beyond this paper. From reading the paper, not all details were completely clear to me. I had to look up Bayesian Optimisation, what was meant with 'batch of candidates', etc. Since the proposed algorithmic improvements are described on a mostly high level, it's very likely I missed some of the details of how everything comes together (and this will be similar for other readers less familiar with the field). Even though the target audience might be more knowledged on this topic, I do recommend considering improvements.


Brief questions:

* Def 2.2 feasible Pareto frontier, should $g(x) = 0$ instead of $\geq 0$? By earlier description, $g(x)$ is always $\geq 0$?

* In caption Figure 1, MW7 problem has d=2 but in Figure 2 it has d=10? I assume this is a typo and should be 10?

## Textual remarks

* "The three right-most plots **shows illustrate**"

* "...that simultaneously maximizes all objectives and satisfies all constraints" -> "...maximizes all $M$ objectives and all $V$ constraints"?

* Def 2.2 $x \in X$ -> $\mathcal{X}$?, same for $x'$? Same for $P(X)$ etc? (cf. Multi-Objective Optimization paragraph uses $x \in \mathcal{X}$).

* Fig 2 caption: "uses an hypervolume acquisition function is known to reward" -> "... that is known to reward"

* p5: SCBO is used without being clear what it stands for, "... (SCBO)" does not occur?


**Q7 Justification For Your Score:**

-Clarity of writing: hard to grasp the complete picture/details for readers less familiar with the field

+experiments include comparison to several other related work; showing strong, convincing results of improving the state-of-the-art.

+Bayesian optimisation, and their proposed algorithm, is very relevant and expected to be of interest to the UAI community.

+broad summary of relevant work (high level descriptions)



**Q9 Complying With Reviewing Instructions:**

1: Yes.

---

### Decision · Program_Chairs · 2022-05-15

**Decision:**

Accept (Oral)

**Comment:**

Meta Review: Both reviewers agree that the paper has merit and deserves to be published. I'm particularly interested in the presentation of this work at the conference as it shows strong performance and addresses high-dimensional problems with the Bayesian optimisation framework. I encourage the authors to address the small issues on clarity and presentation.